# Discovery of a Closterovirus Infecting Jujube Plants Grown at Aksu Area in Xinjiang of China

**DOI:** 10.3390/v15020267

**Published:** 2023-01-17

**Authors:** Qian Lu, Guoping Wang, Zuokun Yang, Yanxiang Wang, Buchen Yang, Jianyu Bai, Ni Hong

**Affiliations:** 1Key Lab of Plant Pathology of Hubei Province, College of Plant Science and Technology, Huazhong Agricultural University, Wuhan 430070, China; 2Laboratory of Fruit Trees Disease, Institute of Economic Forestry, Xinjiang Academy of Forestry Sciences, Urumqi 830063, China

**Keywords:** jujube, persimmon ampelovirus, genomic sequence, RT-PCR

## Abstract

Chinese jujube (*Ziziphus jujuba* Mill.) is a widely grown fruit crop at Aksu in Xinjiang Uygur Autonomous Region of China. Viral disease-like symptoms are common on jujube plants. Here, for the first time, we report a virus tentatively named persimmon ampelovirus jujube isolate (PAmpV-Ju) infecting jujube plants. The virus was identified using high-throughput sequencing from a jujube plant (ID: AKS15) and molecularly related to viruses in the family *Closteroviridae*. The genomic sequences of two PAmpV-Ju variants named AKS15-20 and AKS15-17 were determined by RT-PCR amplifications. The genome structure of PAmpV-Ju was identical to that of a recently reported persimmon ampelovirus (PAmpV) and consisted of seven open reading frames. The genomes of AKS15-20 and AKS15-17 shared 83.7% nt identity with each other, and the highest nt sequence identity of 79% with two variants of PAmpV. The incidence of PAmpV-Ju on Aksu jujube plants was evaluated by RT-PCR assays. The phylogenetic analysis of amplified partial sequences coding for polymerase, HSP70h, and CP revealed two phylogenetic clades represented by AKS15-20 and AKS15-17. Our study provides important evidence for understanding viruses infecting jujube plants and establishing efficient measures to prevent virus spread.

## 1. Introduction

The family *Closteroviridae* contains more than 50 recognized species, of which many plant viruses cause serious diseases of economically important crops [1]. In the International Committee on Taxonomy of Viruses (ICTV) report posted on May 2022 (ICTV Master Species List, 2022v1), viruses in the family *Closteroviridae* are classified into seven genera: *Closterovirus*, *Ampelovirus*, *Crinivirus* and *Velarivirus*, *Olivavirus*, *Bluvavirus*, and *Menthavirus* (https://ictv.global/filebrowser/download/6480, accessed on 20 September 2022). The newly established genus *Olivavirus* includes three recently reported virus species, *Olive leaf yellowing-associated virus*, *Actinidia virus 1*, and *Persimmon virus B* [2,3,4]. The genera *Bluvavirus* and *Menthavirus* contain *Blueberry virus A* and *Mint vein banding-associated virus* [5,6], respectively. More recently, a virus named persimmon ampelovirus (PAmpV) was identified from persimmon plants [7]. PAmpV is phylogenetically related to species in the genus *Ampelovirus*. Viruses in the family *Closteroviridae* have genomes of 13 to 19.3 kb positive-sense, single-stranded (+ss) RNAs [8,9], which are commonly organized with a replication-related module consisting of two open reading frames (ORF) 1a and 1b, and a five-gene module encoding a small hydrophobic protein, a homolog of the plant heat shock proteins HSP70 (HSP70h), a ∼60 kDa protein, a coat protein (CP), and a defined minor coat protein (CPm). Some viruses in the genus *Ampelovirus* do not possess a CPm. Great genetic variability has been found in viruses in the family *Closteroviridae* [10]. For example, the citrus tristeza virus (CTV) population consists of highly divergent molecular variants, which are classified into different genotypes [11,12,13,14]. The sequence variations might be involved in the virus evolution to adapt to different stresses [15,16].

Jujube (*Ziziphus jujuba* Mill.) belongs to the *Ziziphus* genus of the family *Rhamnaceae* [17]. Jujube, which originated from China, has been widely grown in China [18]. In the Xinjiang Uygur Autonomous Region of China in particular, jujube has become one of the most economically important fruit crop species. In recent years, a jujube mosaic disease has been observed in Xinjiang Uygur Autonomous Region. By using high-throughput sequencing (HTS) for small RNA and rRNA-deleted RNA combined with RT-PCR analyses, we identified an emaravirus tentatively named jujube yellow mottle-associated virus (JYMaV) from jujube plants [19]. Meanwhile, from an RNA-seq library we identified two contigs partially matching the genome sequences of viruses in the family *Closteroviridae*, indicating that a virus belonging to the family was present in the RNA-seq sample. Then, the genome sequences of the virus were determined by Sanger sequencing. The virus had the typical genomic structure of viruses in the family *Closteroviridae* and its genome showed high sequence identity with the recently reported PAmpV from persimmon plants [7]. Here, the newly identified virus is tentatively named persimmon ampelovirus jujube isolate (PAmpV-Ju). Furthermore, the incidence and molecular divergence of the virus is investigated.

## 2. Materials and Methods

### 2.1. Virus Source and Sample Collection

The sample AKS15, in which contigs matching sequences of reported closteroviruses were identified by HTS analysis [19], was used for the amplification of the PAmpV-Ju genome. In 2018, leaf samples of 62 jujube plants (Appendix A) were collected from the Aksu area, Xinjiang Uygur Autonomous Region, and tested for the presence of PAmpV-Ju by RT-PCR. Except for two samples, AKS10 and AKS11, the other samples showed viral disease-like symptoms, including semi-transparent chlorotic spots, blotches on young leaves, and leaf malformation. A leaf sample of a healthy jujube plant from Hubei Province was used as a negative control in all RT-PCR assays.

### 2.2. Amplification of the Viral Genome

Primers were designed based on contig17 and contig20 sequences (Figure 1, Appendix A), respectively. The total RNA was extracted from the sample AKS15 using a silica spin column-based RNA isolation method [20]. The resulting RNAs were used as templates for the synthesis of the first strand cDNAs using an M-MLV reverse transcriptase (Promega, Madison, WI, USA) and random hexamer primers pd(N)6 (Takara, Dalian, China). The solutions and conditions of the PCR reactions were similar to those previously described [19], except that the extension time and annealing temperature varied depending on the sizes of the amplicons and primer sets used. The commercial kits for the rapid amplification of cDNA ends (RACE) (Takara, Dalian, China) were used to determine the 5′ and 3′ terminal sequences of the viral genome. The 5′ RACE reaction was performed according to the manufacturer’s protocol. For 3′ RACE, a poly (A) tail was added to the 3′ end of the total RNAs using the poly (A) polymerase kit (Takara, Dalian, China), and cDNA was generated using the 3′CDS (SMARTer Π A Code domain sequence) Primer A provided in the SMARTer RACE 5′/3′ Kit and SMARTScribe Reverse Transcriptase [21]. The primers used for the amplification of the viral genomic RNA segments are listed in Appendix A.

The PCR products were gel-purified and ligated into the pMD18-T vector (Takara, Dalian, China). At least three positive clones of each amplicon were sequenced at Sangon Biological Engineering & Technology and Service Co., Ltd., (Shanghai, China). The obtained sequences were assembled into contiguous sequences by overlapping common regions (>50 bp) of the adjacent amplicons with a sequence identity of >99%.

### 2.3. RT-PCR Detection of PAmpV-Ju

For the efficient detection of the virus, five sets of primers were designed according to the sequences of the viral genome (Table 1). Among these primer sets, CP-F1/CP-R1 and CP-F2/CP-R2 targeting the CP gene of the virus were designed based on the specific sequences of contig17 and contig20, respectively. Other primer sets targeting the polymerase, HSP70h, and CP coding regions were designed based on the conserved sequences of contig17 and contig20. The total RNA extraction and reverse transcription (RT) were conducted as described above. The PCR products were separated by electrophoresis on 1.5% agarose gels, stained with ethidium bromide, visualized under UV light, cloned, and sequenced.

### 2.4. Sequence Analysis

The ORF predictions were carried out using ORFfinder on the NCBI website (https://www.ncbi.nlm.nih.gov/orffinder/, accessed on 20 August 2022). Conserved domains in the predicted proteins were identified using the Conserved Domain Database (CDD) on the NCBI website (https://www.ncbi.nlm.nih.gov/Structure/cdd/docs/cdd_search.html/, accessed on 20 August 2022) [22]. Multiple alignments and identity analyses of nucleotide and amino acid sequences were performed using the ClustalW2 program (https://www.ebi.ac.uk/Tools/psa/lalign/, accessed on 1 September 2022). The used parameters are matrix selection BLOSUM 62, gap open value −1, gap extend value −5, default value of 10, and the output format Markx 0. Phylogenetic trees were constructed using a neighbor-joining method with 1000 bootstrap replications in MEGA 7.0 [23].

## 3. Results

### 3.1. Sequences of PAmpV-Ju Identified by RNA-Seq Analysis

In our previous work, two libraries (SRA ID: PRJNA684042) prepared from the rRNA-deleted RNAs of leaf samples of jujube plants AKS6 and AKS15 were subjected to HTS [19]. The initial BlastN analysis using RNA-seq-derived contigs against the NCBI database showed that the two large contigs with lengths of 14,142 nt (contig17) and 14,229 nt (contig20) matched the genomic sequence of plum bark necrosis stem pitting-associated virus (PBNSPaV), a virus belonging to the family *Closteroviridae*. Contig17 and contig20 showed 64.7% and 64.5% nt identities with the genome sequence (GenBank accession number EF546442) of the PBNSPaV isolate PL186, respectively, indicating the possible presence of a closterovirus. When the genomic sequence of persimmon ampelovirus (PAmpV) was available in NCBI GenBank, the sequence alignment showed that contig17 and contig20 well matched the genomic sequence of PAmpV-PBs3 (accession number LC488185) with 78.2% and 80.4% nt sequence identities, respectively. The two contigs shared 82.7% nt sequence identity with each other. The results indicated that two divergent variants of PAmpV-Ju were present in the sample AKS15.

### 3.2. Amplification and Analysis of the Genomic Sequence of PAmpV-Ju

The full-length genome of PAmpV-Ju was determined by RT-PCR and 5′ and 3′ RACE reactions using two sets of primers designed according to the contig17 and contig20 sequences, respectively (Appendix A). The sequencing results showed that the clones of each amplified fragment had over 99% nt identity. Then, one clone sequence of each fragment was used for sequence assembling of the virus genome. Eventually, two full-length genome sequences of PAmpV-Ju variants AKS15-17 and AKS15-20, which were named according to their original contig numbers, were obtained from the sample AKS15. The genome sequences of the two variants were submitted to NCBI GenBank with the accession numbers OP673404 and OP673405, respectively. Variants AKS15-17 and AKS15-20 had the same genome size of 14,209 nt, and shared 99.1% and 99.2% nt identity with contig17 and contig20, respectively, indicating that the contig sequences assembled from RNA-seq data were reliable.

The sequences of the amplified fragments were individually compared with the sequences of contig17 and contig20. It was noticed that the overlapping region of fragments F17-4 and F17-5 was 86 nt longer than the corresponding sequence of contig17, and the fragment F20-4 was 40 nt longer than the corresponding sequence of contig20. The 3′ UTR in the assembled contig20 was 80 nt more than the sequence obtained by the 3’ RACE reaction. The results indicated that assembling errors might have occurred in these contig sequences.

### 3.3. Characterization of PAmpV-Ju Genome

PAmpV-Ju variants AKS15-17 and AKS15-20 had the same genomic structure consisting of seven ORFs and two UTRs at the 5′ and 3′ termini (Figure 1). The two variants shared 83.7% nt sequence identity for their full genomes, and 85.5% (ORF1a) −98.3% (ORF2) aa identities for their ORFs. The genomes of two PAmpV-Ju variants had the highest sequence identity of 79.1% with two variants of PAmpV, and 67.6% sequence identity with the representative isolate PL186 of PBNSPaV (Table 2). The ORF and UTR sizes of PAmpV-Ju were within the ranges of the corresponding regions of viruses in the family *Closteroviridae* [9]. ORFs 2–6 of the two PAmpV-Ju variants AKS15-17 and AKS15-20 shared the same size as that of PAmpV, but the 5′ UTR and ORF1a of the PAmpV-Ju variants were 16 nt longer and 69 nt shorter than that of PAmpV, respectively. Multiple alignment for the aa sequences of ORF1a of the PAmpV-Ju variants AKS15-17 and AKS15-20, and PAmpV-PBs3 showed that the N-terminal of ORF1a was variable with PAmpV-Ju having 23 aa deletions at sites aa 145–168 (Figure 2A). In addition, AKS15-17 and AKS15-20 had one aa (D) insert in their ORF1b (Figure 2B).

The ORF1a of PAmpV-Ju variants AKS15-17 and AKS15-20 encodes a polyprotein of 2333 amino acids with a molecular mass (Mr) of 260.2 kDa. The Pfam search identified a methyltransferase motif (MTR, cl03298) at aa 789–1115 and a helicase motif (HEL, cl26263) at aa 2035–2299, which are necessary for virus replication. Analysis using the CDD tool identified a conserved domain of an 2OG-Fe (II) oxygenase superfamily (2OG-FeII_Oxy_2, cl21496) at position aa 338–431. Further analysis showed that the domain was also present in the corresponding proteins of the PAmpV isolate PBs3, PBNSPaV, and some other viruses in the family *Closteroviridae*. The phylogenetic analysis for the 2OG-FeII_Oxy_2 sequences revealed that PAmpV-Ju and PAmpV formed a clade and separated from PBNSPaV and GLRaV3 (Appendix A).

ORF 1b is expressed by a +1 ribosomal frameshift [24]. The Pfam search found an RdRp motif at aa 53–496 (cl03049). ORF2 encodes a protein consisting of 59 aa with a Mr of 6.3 kDa (P6). ORF3 and ORF4 encode HSP70h and HSP90h, which contain an HSP70h (cd10170) motif and an HSP90h (cl20248) homolog motif at sites aa 3–360 and aa 22–497, respectively. ORF5 and ORF6 encoded a CP and a CPm with Mr 35.6 kDa and 24.3 kDa, respectively. Pfam search found a closterovirus CP (cl03354) motif at aa 209–308 of CP.

When the aa sequences of RdRp, HSP70h, and CP of the PAmpV-Ju variants AKS15-17 and AKS15-20, PAmpV isolates PBs3 and PBv2s2, and the representative viruses in the family *Closteroviridae* available in NCBI GenBank were subjected to phylogenetic analyses, AKS15-17 and AKS15-20 always clustered together and were closely related to two variants of PAmpV. These four variants formed a clade, which distantly related to the clade consisting of PBNSPaV isolates (Figure 3).

### 3.4. RT-PCR Detection of PAmpV-Ju in Jujube Plants

To understand the incidence of PAmpV-Ju, the leaf samples collected from 62 jujube plants (Appendix A) grown at Aksu area in Xinjiang Uygur Autonomous Region of China were subjected to RT-PCR assays for PAmpV-Ju using five sets of primers (Table 1). The results showed that the detection efficiency differed depending on the primer sets. Four, five, two, and four samples were positive for PAmpV-Ju as tested using primer sets Pol-F/ Pol-R, HSP70-F/HSP70-R, CP-F1/CP-R1, and CP-F2/CP-R2, respectively. Of these positive samples, two samples, AKS15 and AKS19, were identified to be PAmpV-Ju-positive using five primer sets. To improve the detection efficiency, the second round of PCR amplifications were carried by using the products from CP-F1/CP-R1 and CP-F2/CP-R2 as templates and a primer set CP-F/CP-R as inner primers. Then, 12 and 15 samples were found positive for PAmpV-Ju, respectively. Of these samples, five samples were positive for the virus as detected using the products from both CP-F1/CP-R1 and CP-F2/CP-R2 as templates. The other positive samples were identified by using templates produced from primer set CP-F1/CP-R1 or CP-F2/CP-R2. Taking together, of the 62 samples evaluated using the nest primer CP-F/CP-R, 22 samples were positive for PAmpV-Ju, accounting for 35.5%. The PCR products from 4, 5, and 22 samples obtained using primer sets Pol-F/ Pol-R, HSP70-F/HSP70-R, and CP-F/CP-R, respectively, were cloned and sequenced. The sequences were submitted to GenBank with accession numbers OP709436–OP709449 for partial coding sequences of polymerase, OP709422–OP709435 for partial coding sequences of HSP70h, and OP709450–OP709514 for partial coding sequences of CP, respectively.

The obtained PAmpV-Ju polymerase sequences shared 79.6–99.7% nt and 84.4–99.0% aa identities with each other, and 71.0–72.8% nt and 74.5–76.2% aa identities with the corresponding sequences of PAmpV isolate PBs3 available at NCBI GenBank. The obtained HSP70h sequences shared 86.6–99.8% nt and 84.4–99.0% aa identities with each other, and 71.0–79.5% nt and 94.3–96.6% aa identities with the corresponding sequences of PBv2s2 available at NCBI GenBank. The sequences of 65 CP clones from 22 samples shared 84.9–99.4% nt and 95.2–98.4% aa identities.

These sequences together with the corresponding sequences of PAmpV isolates PBs3 and PBv2s2, and PBNSPaV isolate PL186 referred from the GenBank database were used for the phylogenetic analysis. In the phylogenetic trees, PAmpV-Ju variants from Aksu jujube samples clustered as two clades represented by AKS15-17 and AKS15-20, respectively. The two clades were distanced from PAmpV and PBNSPaV (Figure 4). Notably, clones from samples AKS15, AKS19, AKS5, AKS14, and AKS18 were sequence divergent and distributed in different clades. In the HSP70h sequence-based tree, clones from sample AKS19 clustered into two clades. In the CP sequence-based tree, clones from each of the samples AKS5, AKS14, AKS15, and AKS18 also clustered into two clades. The results suggested the molecular diversity within the Aksu PAmpV-Ju population.

## 4. Discussion

Next-generation sequencing combined with bioinformatic analyses has been widely used for the rapid discovery and characterization of known or novel viruses infecting plants [25,26,27,28,29,30,31,32,33]. In recent years, two new viruses respectively belonging to the families *Fimoviridae* [34] and *Caulimoviridae* [35] have been identified from jujube using HTS technologies. For the first time, this study revealed the natural infection of a virus belonging to the family *Closteroviridae* in jujube plants. RNA-seq analyses revealed that two large contig sequences from a *Z*. *jujuba* cv. ‘Junzao’ plant (ID: AKS15) well matched the genomes of viruses in the family *Closteroviridae*. Consistent with the RNA-seq analyses, Sanger sequencing for the RT-PCR products obtained by using primers designed based on the two contig sequences revealed two divergent variants, AKS15-17 and AKS15-20, of PAmpV-Ju in the sample AKS15. Sequence comparison showed that the two contig sequences covered over 95% of the viral genome and had about 98% nt identity with the sequences derived from Sanger sequencing, indicating that the RNA-seq analysis and sequence assembly was efficient for the genomic characterization of the virus.

The PAmpV-Ju variants AKS15-17 and AKS15-20 had the same genomic size of 14,209 nt and genomic structure consisting of seven ORFs, which is similar to that of a recently reported PAmpV from persimmon plants [7]. However, the 5′ UTR, ORF1a, and ORF1b of PAmpV-Ju were 16 nt more, 69 nt less, and 3 nt more than that of PAmpV, respectively. The analyses for the two complete genomic sequences of variants AKS15-17 and AKS15-20, and the complete and near complete genomic sequences of PAmpV variants PBs3 and PBv2s2, revealed relatively high genetic divergence in the 5′ UTR and the ORF1a, with divergency near 15–20% at the nt level. The ORF1a and ORF1b size fluctuation has also been found in some viruses in the family *Closteroviridae* [36,37,38]. The ORF-by-ORF comparisons for the closely related viruses PAmpV-Ju, PAmpV, and PBNSPaV revealed that they have the same sizes for their ORF3 and ORF4, which code an HSP70h and an HSP90h. These two proteins have been considered to interact directly with the replication complex to increase the replication of the virus and form complexes to facilitate viruses to enter the cell [39].

For PAmpV-Ju and PAmpV, the aa divergence values of ~10% for their RdRp, HSP70h, and CP are much less than the species-discriminating threshold of 25% approved by the International Committee of Taxonomy of Viruses. In the phylogenetic trees based on the aa sequences of proteins RdRp, HSP70h, and CP of fully sequenced PAmpV variants AKS15-17 and AKS15-20 infecting jujube, and variants PBs3 and PBv2s2 infecting persimmon, and representative viruses in the family *Closteroviridae*, four PAmpV variants always clustered into the same clade and are distantly related to PBNSPaV. However, when the partial nucleotide sequences of genes coding for the Pol, HSP70h, and CP of PAmpV-Ju amplified from field jujube samples were analyzed, the PAmpV-Ju variants clustered in a large clade and separated from PAmpV variants PBs3, PBv2s2, and PBNSPaV, indicating that the PAmpV variants infecting jujube were molecularly different from that infecting persimmon. Whether the molecular divergence of the virus variants is relative to their host selection remains a topic for further evaluation. Since the PAmpV-Ju and PAmpV variants PBs3 and PBv2s2 have a close relationship and a common genome organization with the virus PBNSPaV in the genus *Ampelovirus* [9], we contend that PAmpV-Ju and PAmpV are two divergent isolates of a novel virus in the genus *Ampelovirus*. The reported closteroviruses have complicated populations consisting of highly variable variants, with the highest divergency of some proteins up to more than 25%, such as AcV-1 [38], GLRaV-3, GLRaV-4, and PeVB [4]. The PAmpV-Ju variants tested in this study showed a divergency of less than 10%. However, these variants could be clearly divided into two groups represented by variants AKS15-17 and AKS15-20, as seen in the trees based on the partial nucleotide sequences of Pol, HSP70h, and CP of the virus. The results showed the molecular variability of the virus and further supported the reliability of the viral sequences determined by RNA-seq analysis.

Viruses can easily be transmitted among perennial woody plants through vegetative propagation. Jujube plants are commonly propagated by grafting on rootstocks. The frequent multiple artificial manipulations of infected scion and rootstocks might result in the wide transmission of the virus and the mixed infection of viruses. Most plant viruses in the family *Closteroviridae* are transmitted by hemipteran insect vectors, such as aphids (Aphididae) or whiteflies (Aleyrodidae) [40,41,42]. Artificial manipulations and potential vector transmission for the virus might contribute to the mixed infection of diverse variants of PAmpV-Ju and its co-infection with other viruses in a single jujube plant, as has been found for some viruses in the family *Closteroviridae* [43,44]. The RT-PCR analyses for PAmpV-Ju revealed a high occurrence frequency of PAmpV-Ju in Xinjiang, indicating a long-term infection of the virus in jujube plants and the presence of potential vectors. The distribution of PAmpV-Ju in the areas outside Xinjiang needs further investigation. Previously, three diseased jujube samples were subjected to HTS analyses for viruses [19]. PAmpV-Ju was only detected in the sample AKS15. All jujube samples tested by RT-PCR analyses for PAmpV-Ju in this study were previously tested for the presence of JYMaV [19], which revealed a robust association between JYMaV and the jujube chlorotic leaf spot and mottle disease occurring at Aksu. Of the 23 samples found positive for PAmpV-Ju, two samples were asymptomatic, indicating a latent infection of PAmpV-Ju. We also found that except for two symptomatic samples negative for JYMaV, the other samples were symptomatic and mix-infected with PAmpV-Ju and JYMaV. In this case, it is not possible to evaluate the association of the virus with jujube disease. The mixed infection of different viruses in plants is very common and may affect the plant symptom development [45]. Since the samples tested in this study had limited origins and the potential for having been infected by other untested viruses in these samples, we cannot conclude the effect of the mixed infection of PAmpV-Ju with JYMaV on the symptom severity of jujube plants at present. In addition, out of the 23 PAmpV-Ju-positive samples, only two samples, AKS15 and AKS19, were identified to be PAmpV-Ju-positive using five primer sets. The molecular variation of the virus might result in the failure of RT-PCR and lead to false negative results, as was found for some other viruses [46,47,48,49]. The low titer or uneven distribution of the virus might also affect the detection [50].

## Figures and Tables

**Figure 1 viruses-15-00267-f001:**
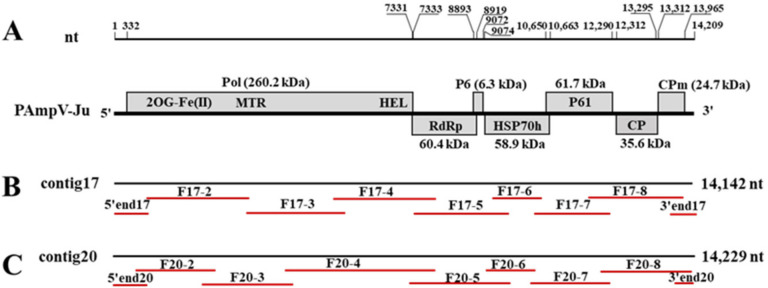
Genome organization of persimmon ampelovirus jujube isolate (PAmpV-ju) showing relative positions of the open reading frames (ORFs) and their expression products (**A**), and the positions fragments amplified by RT-PCR on contig17 and contig20 (**B**,**C**). The red lines under two contigs represent the fragments amplified using primer sets specific for contig17 and contig20.

**Figure 2 viruses-15-00267-f002:**
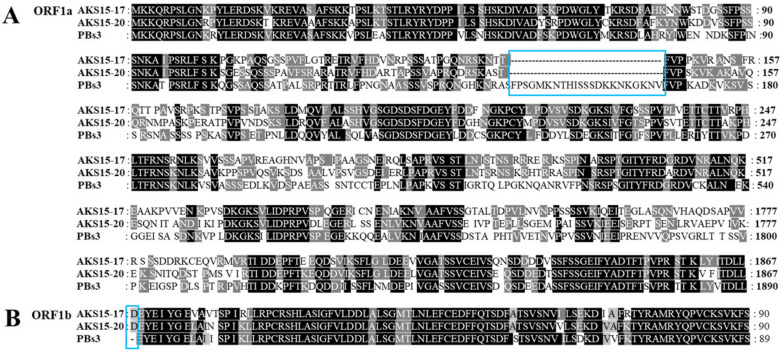
Multiple alignment of partial amino acid sequences of ORF1a (**A**) and ORF1b (**B**) of PAmpV-Ju variants AKS15-17, AKS15-20, and PAmpV isolate PBs3. The deletions are indicated by blue boxes.

**Figure 3 viruses-15-00267-f003:**
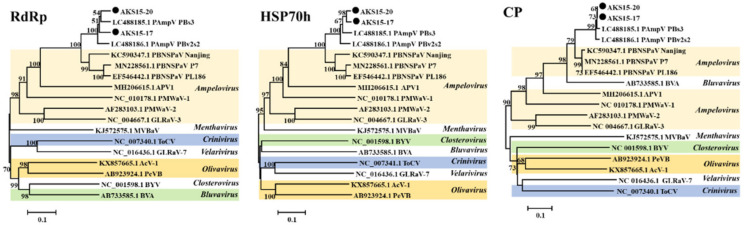
Phylogenetic analysis of PAmpV-Ju variants AKS15-17 and AKS15-20, PAmpV and representative members of the family *Closteroviridae* based on the amino acid sequences of their RdRp, HSP70h, and CP. The phylogenetic trees were constructed using a neighbor-joining algorithm with 1000 bootstrap replications; bootstrap support values >50% are shown at the nodes. PAmpV, persimmon ampelovirus; PBNSPaV, plum bark necrosis stem pitting-associated virus; APV1, air potato virus 1; PMWaV-1, pineapple mealybug wilt-associated virus 1; PMWaV-2, pineapple mealybug wilt-associated virus 2; GLRaV-3, grapevine leafroll-associated virus 3; MVBaV, mint vein banding-associated virus; ToCV, tomato chlorosis virus; GLRaV-7, grapevine leafroll-associated virus 7; AcV-1, actinidia virus 1; PeVB, persimmon virus B; BYV, beet yellows virus; BVA, blueberry virus A.

**Figure 4 viruses-15-00267-f004:**
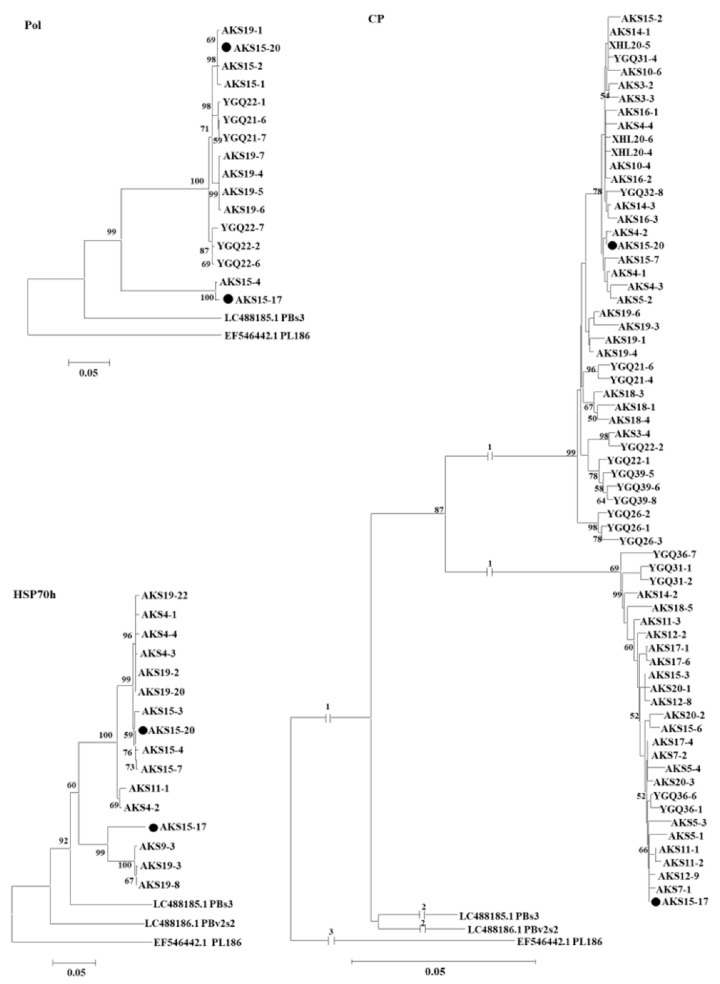
Phylogenetic analysis of PAmpV-Ju variants based on the nucleotide sequences amplified using the primer sets Pol-F/Pol-R, HSP70-F/HSP70-R, and CP-F/CP-R. Each variant name is followed by clone numbers. The phylogenetic trees were constructed using a neighbor-joining algorithm with 1000 bootstrap replications. Bootstrap support values >50% are shown at the nodes. The two variants AKS15-17 and AKS15-20 of PAmpV-Ju were marked with black dots. PBs3 and PBv2s2 were two isolates of persimmon ampelovirus. PL186 was an isolate of plum bark necrosis stem pitting-associated virus.

**Table 1 viruses-15-00267-t001:** Primers designed for the detection of PAmpV-Ju.

Primer Name	Sequence ^a^ (5′-3′)	Location (nt) ^b^	Size (bp)	Target Sequence
Pol-F	AAACTCAAGGGTCGTAGAGC	1220–2162	942	contig17; contig20
Pol-R	CGCCCATCAACGTGTTTCATCCA			
HSP70-F	GGCTAARGGTTGTACTGAAGAGTC	9097–9628	531	contig17; contig20
HSP70-R	GTACCTCCTCCAAAATCATAGAC			
CP-F1	TACCGAGGCATACATGTTA	11,933–13,322	1389	contig17
CP-R1	CGCCTAATCGCAACGTCAC			
CP-F2	TCCACTCGTATTATCGCATGA	12,018–13,340	1322	contig20
CP-R2	TGTACCCATTACTCCGCTTCAC			
CP-F	GATGARTGGGTACCYGGGGATT	12,357–13,334	977	contig17; contig20
CP-R	CGCGCAAKACAACAGAAGACAT			

^a^ R: A/G; Y: C/T; K: G/T. ^b^ The locations refer to nucleotide positions corresponding to the genome sequence of PAmpV-Ju determined in this study.

**Table 2 viruses-15-00267-t002:** Genome sequence comparison of PAmpV-Ju variants AKS15-17 and AKS15-20 infecting a jujube plant AKS15 with PAmpV isolates PBs3 and PBv2s2, and PBNSPaV isolates PL186 and Nanjing.

Virus	Isolate (Variant) ^a^	Genome		ORF1a			ORF1b			ORF2		
				(Polyprotein)			(RdRp)			(P6)		
		nt	nt%	nt	nt%	aa%	nt	nt%	aa%	nt	nt%	aa%
**PAmpV**	**AKS15** **-17**	**14,209**		**7002**			**1587**			**180**		
	**AKS15-20**	**14,209**	**83.7**	**7002**	**80.1**	**85.5**	**1587**	**86.2**	**93.2**	**180**	**87.2**	**98.3**
PAmpV	PBs3	14,262	79.1/79.0	7071	74.5/75.4	79.1/79.5	1584	80.3/83.0	91.3/93.2	180	75.6/80.9	89.8/91.5
	PBv2s2	7070	-	-	-	-	1584	80.7/80.8	90.2/90.1	180	79.0/80.7	88.1/89.8
PBNSPaV	PL186	14,214	67.6/67.4	7032	64.8/64.6	55.7/55.7	1575	73.0/72.7	77.5/78.1	174	65.8/65.4	64.3/64.3
	Nanjing	14,234	67.0/67.1	7035	61.8/55.8	33.3/36.7	1578	72.6/73.0	79.7/78.9	174	55.9/66.0	60.7/60.7
**ORF3**			**ORF4**			**ORF5**				**ORF6**		
**(HSP70h)**			**(P61)**			**(CP)**				**(CPm)**		
**nt**	**nt%**	**aa%**	**nt**	**nt%**	**aa%**	**nt**	**nt%**	**aa%**		**nt**	**nt%**	**aa%**
**1590**			**1641**			**984**				**654**		
**1590**	**85.9**	**95.8**	**1641**	**84.6**	**96.3**	**984**	**85.9**	**96.3**		**654**	**92.4**	**93.5**
1590	82.6/81.2	93.2/92.6	1641	82.9/81.8	94.3/92.7	984	83.9/83.1	93.3/92.0		654	87.2/88.1	88.9/90.8
1590	79.3/78.9	92.2/92.1	1641	79.8/78.8	88.8/89.0	984	81.4/82.7	93.3/91.7		654	86.5/87.8	86.6/88.9
1590	71.4/70.8	76.9/77.3	1641	70.3/70.3	75.8/75.5	978	68.1/68.2	69.3/70.2		672	66.3/66.7	58.3/60.7
1590	70.0/70.8	78.4/78.4	1641	70.6/70.5	73.8/73.8	978	68.3/69.4	69.3/70.5		657	63.5/63.8	57.0/56.9

^a^ The sequences determined in this study are in bold.

## Data Availability

The data presented in this study are available in article and Appendix A. The Raw fastq files of the RNA-Seq library of AKS-15 was available in the GenBank with an SRA number PRJNA684042.

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
