# Peer review of "Discovery of a Closterovirus Infecting Jujube Plants Grown at Aksu Area in Xinjiang of China"

_viruses, 2023, doi:10.3390/v15020267_

Round 1
Author Response
Point1: As follows from the reference list, the authors discovered several viruses infecting fruit trees, and jujube in particular. However, molecular characteristics were obtained in silico for the “paper” virus. They need to be confirmed and clarified for the isolated virus.
Response: We thank the reviewer for the evaluation. We agree with the reviewer that more biological studies are necessary to characte the virus. It has been discussed.
Point2: Aforementioned 5 pairs of primers were used for the RT-PCR molecular diagnostics of 62 jujube samples from the Xinjiang Uygur Autonomous Region to analyze an Incidence of the closterovirus infection (Supplementary table). The infection was 100% found for two jujube samples, in which the virus was in silico discovered. Infection of other 29% of samples was diagnosed with 1, 2 or 3 pairs of primers. This can be explained by a great potential for genetic variation, rapid evolution and adaptation of RNA viruses and the lack of activity of reverse transcriptase for the 3'-->5' exonuclease proofreading. It means that for the mass diagnostics analysis in practice it is necessary either to reduce the cost of the diagnostic RT-PCR analysis, or in addition to use an independent method.
Response: Yes, only two positive samples AKS15 and AKS19 were commonly identified by using five primer sets. In AKS15, the virus was in silico discovered. Our detection promers were designed based on the two genomice sequences derived from AKS15. The virus is molecularly variable, which might result in the defaut using primers targeting to different sites. For the reliable diagnostic analysis, it is necessary to design primers basing on more molecular data and develop other methods. This is discussed in the revised manuscript.
Point 3: I think number 1 should be avoided in the manuscript title. A reader meets easily the closterovirus Number 1 through the text.
Response: The title has been revised according the reviewer’s suggestion.
Point 4: Line 84 - CDS abbreviation should be deciphered.
Response: The CDS has been deciphered as SMARTer Π A Code domain sequence.
Point 5:Line 142 – correct “that assembling errs might occurred in these fragments.”
Response: It has been changed as “ that assembling errs might occurred in these contig sequences”.

Reviewer 2 Report
In this manuscript, a virus with close relationship to the members of family Closteroviridae was identified through high-throughput sequencing (HTS) from a jujube plant (ID: AKS15) in Xinjiang Uygur Autonomous Region of China. This virus was tentatively named as jujube closterovirus 1 (JuCV-1). Authors then determined genomic sequences of two JuCV-1 variants (AKS15-20 and AKS15-17) by RT-PCR amplification and 3’/ 5’ RACE and characterized the genome structure of them. Multiple alignments, identity analyses, and phylogenetic analysis were performed to illustrate the relationship between JuCV-1 and other members of family Closteroviridae. Authors also investigated the incidence of the JuCV-1 on Aksu jujube plants by RT-PCR assays.
In general, this manuscript needs major revision before publication. Some parts are incomprehensible or difficult to understand, the English writing needs proofreading and polishing from a native speaker. Following are some general comments to make this manuscript stronger and some writing suggestions that will help improve this manuscript:
Comment 1:In the newest report of International Committee of Taxonomy of Viruses (ICTV), the criteria demarcating species in the genus Ampelovirus are: 1) Particle size. 2) Size of CP, as determined by deduced amino acid sequence data. 3) Serological specificity using discriminatory monoclonal or polyclonal antibodies. 4) Genome structure and organization (number, relative location and size of the ORFs). 5) Amino acid sequence of relevant gene products (RdRP, CP, HSP70h) differing by more than 25%. 6) Vector species and specificity. 7) Magnitude and specificity of natural host range. 8) Cytopathological features (i.e., aspect of inclusion bodies and origin of cytoplasmic vesicles). However, as mentioned in the manuscript, the amino acid sequences of RdRP, CP, HSP70h are differing from persimmon ampelovirus (PAmpV) by less than 10%. This failed to reach the species discriminating threshold of 25 % approved ICTV, so I won’t suggest authors to introduce a new name for this. Instead, PAmpV AKS isolate or PAmpV Jujube variant will be better ways to name it.
Comment 2: The manuscript can be improved by providing some studies on the biological features of this virus (virus inoculation and symptom observation, host range detection, particle size measurement, serological specificity analysis, or cytopathological features analysis etc. )
Comment 3: This manuscript lacks experiments fulfilling Koch's postulates, making it hard to draw any conclusion about the link between the jujube mosaic disease and the virus identified in this manuscript. It will largely decrease the importance of this finding.
Comment 4: In “Materials and Methods” section, part of description in “2.1. Virus Source” should be moved to “Results 3.1 Sequences of JuCV-1 Identified by RNA-Seq Analysis”.
Comment 5:For result “3.2 Amplification and Analysis of Genomic Sequence of JuCV-1”, the description are incomprehensible. I recommend authors to rewrite this part and illustrate the cloning strategy with the help of a figure.
Comment 6:Authors didn’t mention “Table 2” in any part of manuscript. All the Figures and Tables should be properly cited in the manuscript where they were used to illustrate data.
Line 21: Change “the virus” to “JuCV-1”
Line 23: Delete “distantly”
Line 39: Change “the virus” to “PAmpV”
Lines 40- 46: The sentence is too long to read. Please revise it.
Lines 49: Could add this reference (Liu Q, Zhang S, Mei S, Zhou Y, Wang J, et al. (2021) Viromics unveils extraordinary genetic diversity of the family Closteroviridae in wild citrus. PLOS Pathogens 17(7): e1009751.)
Line 52: Change “Jujube originated from China and has been widely grown in China” to “Jujube which originated from China has been widely grown in China”
Line 95: Delete “two primer sets”
Line 98: Delete “to”
Lines 104-105: Delete “To have a view for the molecular characteristics of the virus”
Lines 121- 125: The sentence is too long to read. Please revise it.
Line 134: “Sequencing results showed that clones of each product had over 99% nt identity.” Don’t quite understand. Please revise it.
Lines 158-161: Change “Each of ORFs 2-6 of variants AKS15-17 and AKS15-20 had the same size with that of PAmpV, but the 5’ UTR and ORF1a of JuCV-1 variants were respectively 16 nt more and 69 nt less than that of PAmpV. ” to “ORFs 2-6 of two JuCV-1 variants share the same size with that of PAmpV, but the 5’ UTR and ORF1a of JuCV-1 variants are 16 nt longer and 69 nt shorter than that of PAmpV respectively”
Line 162: Change “145-168 aa” to “aa 145-168”
Line 167: Change “The deletion positions were boxed.” to “The deletions were indicated by blue boxes.”
Line 173: Change “338-431 aa” to “aa 338-431”
Line 181: Change “3-360 aa and 22-497 aa” to “aa 3-360 and aa 22-497”
Line 208: Change “RT-PCR assays for JuCV-1” to “RT-PCR assays for detecting JuCV-1”
Line 213: “outer primer” should be “inner primer”?
Line 214 and 215: Change “showed positive reaction” “were positive for JuCV-1”
Line 303: Change “an old infection of the virus” to “a long-term infection of the virus”
Line 304: Change “reminds for further investigation” to “needs further investigation”
Line 246-313: The whole “Discussion” section needs revision.
Author Response
Point 1: In general, this manuscript needs major revision before publication. Some parts are incomprehensible or difficult to understand, the English writing needs proofreading and polishing from a native speaker.
Response: We thank the reviewer for the evaluation and suggestive comments. We have revised the manuscript according to the reviewer’s comments.
Point 2: In the newest report of International Committee of Taxonomy of Viruses (ICTV), the criteria demarcating species in the genus Ampelovirus are: 1) Particle size. 2) Size of CP, as determined by deduced amino acid sequence data. 3) Serological specificity using discriminatory monoclonal or polyclonal antibodies. 4) Genome structure and organization (number, relative location and size of the ORFs). 5) Amino acid sequence of relevant gene products (RdRP, CP, HSP70h) differing by more than 25%. 6) Vector species and specificity. 7) Magnitude and specificity of natural host range. 8) Cytopathological features (i.e., aspect of inclusion bodies and origin of cytoplasmic vesicles). However, as mentioned in the manuscript, the amino acid sequences of RdRP, CP, HSP70h are differing from persimmon ampelovirus (PAmpV) by less than 10%. This failed to reach the species discriminating threshold of 25 % approved ICTV, so I won’t suggest authors to introduce a new name for this. Instead, PAmpV AKS isolate or PAmpV Jujube variant will be better ways to name it.
Response: Yes, as mentioned in the manuscript, the amino acid sequences of RdRP, CP, HSP70h are differing from persimmon ampelovirus (PAmpV) by less than 10% , which are lower than species discriminating threshold of 25 % approved ICTV. In our original manuscript, considering that PAmpV has not been approved by ICTV, we named our newly identified virus as JuCV-1. However, we discussed that PAmpV and JuCV-1 might be two variants of a new virus. According to the reviewer’s suggestion, we named the virus identified from jujube as PAmpV-Ju.
Point 3: The manuscript can be improved by providing some studies on the biological features of this virus (virus inoculation and symptom observation, host range detection, particle size measurement, serological specificity analysis, or cytopathological features analysis etc. ).
Response: We agree with the reviewer comment. It is important to understand the biological features of this virus. However, at this moment we report the dicovery of the virus, majorly focused on the molecular features of the virus, which is molecularly related to PAmpV, but the two viruses are found in two host species and also show clear genetic distantance as shown in phylogenetic trees. So we provide new information for the virus. Of course, it is necessary to do more studies for the virus in the future. We discussed these points.
Point 3: This manuscript lacks experiments fulfilling Koch's postulates, making it hard to draw any conclusion about the link between the jujube mosaic disease and the virus identified in this manuscript. It will largely decrease the importance of this finding.
Response: For many plant virus, especially woody plant viruses, it is difficult to do experiments fulfilling Koch's postulate due to the difficulty in virus isolation and re-inoculation. We did RT-PCR detection for the virus in diseased jujube plants, but did not find the association of the virus with jujube diseases since some diseased samples were negative for the virus and most samples are positive for JYMaV as reported previously.
Point 4:In “Materials and Methods” section, part of description in “2.1. Virus Source” should be moved to “Results 3.1 Sequences of JuCV-1 Identified by RNA-Seq Analysis”.
Response: Thank you for your suggestion. In the revised manuscript, we indicated the sample used for genome amplification in the section “Materials and Methods”, and the description for contigs was moved to Results 3.1.
Point 5:For result “3.2 Amplification and Analysis of Genomic Sequence of JuCV-1”, the description are incomprehensible. I recommend authors to rewrite this part and illustrate the cloning strategy with the help of a figure.
Response: We have revised the paragragh, and a figure has been added to illustrate the cloning strategy and the genome structure of the virus.
Point 6:Authors didn’t mention “Table 2” in any part of manuscript. All the Figures and Tables should be properly cited in the manuscript where they were used to illustrate data.
Response: All the Figures and Tables have been cited in the revised manuscript.
Point 7:The whole “Discussion” section needs revision.
Response: The “Discussion” section has been revised by emphasis the molecular characteristic and variation of the virus, which are divergent from PAmpV-Ju.
In addition, all other comments for writing have been accepted in the revised manuscript. We thank the reviewer for the detailed evaluation for our manuscript.

Reviewer 3 Report
This study described discovery of a novel ampelovirus infected jujube in China which authors named as jujube closterovirus 1. Within the updated ICTV taxonomy criteria, this jujube closterovirus 1 should be assigned to the species of Persimmon ampelovirus since the amino acid sequence of RdRP, CP and HSP70h differing not more than 25% with Persimmon ampelovirus.
(https://ictv.global/report/chapter/closteroviridae/closteroviridae),
Reporting a new host of Persimmon ampelovirus, this study would benefit from providing data on: i) the detailed information of the virus source samples and process on catch jujube closterovirus contigs in Materials and Methods; ii) the symptoms of positive jujube samples in filed surveys; iii) mixed infection rate of previous jujube emaravirus JYMaV and novel jujube virus; iv) whether mixed infection caused more severe syptomes on jujube plants; v) Genomic organization comparisons diagram should be provided.
Author Response
Point 1: This study described discovery of a novel ampelovirus infected jujube in China which authors named as jujube closterovirus 1. Within the updated ICTV taxonomy criteria, this jujube closterovirus 1 should be assigned to the species of Persimmon ampelovirus since the amino acid sequence of RdRP, CP and HSP70h differing not more than 25% with Persimmon ampelovirus.
(https://ictv.global/report/chapter/closteroviridae/closteroviridae)
Response: It is true, as mentioned in the manuscript, that the amino acid sequences of RdRP, CP, HSP70h are differing from persimmon ampelovirus (PAmpV) by less than 10% , which are lower than species discriminating threshold of 25 % approved ICTV. In our original manuscript, considering that PAmpV has not been approved by ICTV, we named our newly identified virus as JuCV-1. However, we discussed that PAmpV and JuCV-1 might be two variants of a new virus. According to the reviewer’s suggestion, we named the virus identified from jujube as PAmpV isolate Jujube (PAmpV-Ju) in the revised manuscript.
Point 2: Reporting a new host of Persimmon ampelovirus, this study would benefit from providing data on: i) the detailed information of the virus source samples and process on catch jujube closterovirus contigs in Materials and Methods; ii) the symptoms of positive jujube samples in filed surveys; iii) mixed infection rate of previous jujube emaravirus JYMaV and novel jujube virus; iv) whether mixed infection caused more severe syptomes on jujube plants; v) Genomic organization comparisons diagram should be provided.
Response: The detailed information (including origin and symptom) of the virus source samples has been provided in the section “materials”. The method for RNA-Seq data analysis has been reported in previous paper, so that we only give a brief description and provede the reference. To make it be easy understand, the process on catch jujube closterovirus contigs was added in the revised manuscript. The mixed infection of this virus with previous jujube emaravirus JYMaV has been detailed in the section “RT-PCR Detection of PAmpV-Ju in Jujube Plants” and table S2. The effect of mixed infection on syptomes has been discussed. Genomic organization diagram of PAmpV-Ju has be provided , and its comparison with that of PAmpV-PBs3 was illustrated in the section 3.3 .
